# Spirulina (*Arthrospira platensis*): Antiallergic Agent or Hidden Allergen? A Literature Review

**DOI:** 10.3390/foods13071052

**Published:** 2024-03-29

**Authors:** Weronika Gromek, Natalia Kołdej, Marcin Kurowski, Emilia Majsiak

**Affiliations:** 1Polish-Ukrainian Foundation of Medicine Development, Nałęczowska 14, 20-701 Lublin, Poland; 2Student Scientific Association for Allergy, Asthma, and Immunology at the Department of Immunology, Rheumatology, and Allergy Clinic, Medical University of Lodz, 90-419 Lodz, Poland; 3Department of Immunology and Allergy, Medical University of Lodz, 90-419 Lodz, Poland; 4Department of Health Promotion, Faculty of Health of Sciences, Medical University of Lublin, Staszica 4/6, 20-081 Lublin, Poland

**Keywords:** allergy to spirulina, allergy to novel foods, spirulina, microalgae, novel foods

## Abstract

Presently, there has been an increase in the consumption of the blue–green microalga–spirulina (*Arthrospira* species), which dominates 99.5% of the total world production of microalgae. Primarily sold as a dietary supplement, it is also incorporated into snacks, pasta, cookies, and bread. Owing to its nutrient abundance, spirulina has a variety of potential applications. Extensive studies have been conducted on the health benefits of spirulina, but its safety in terms of allergy has received limited attention. Therefore, to bridge this knowledge deficit, this review aimed to evaluate the allergenic and antiallergic potential of spirulina. In the PubMed and Scopus databases using words related to allergy, we attempted to detect papers on hypersensitivity to spirulina. A total of 128 records were identified, of which 49 were screened. Ultimately, in this review, we analyzed four case studies, encompassing a total of five patients with allergies to spirulina. We assessed the severity of allergic reactions following World Allergy Organization (WAO) Anaphylaxis Guidance 2020, which varied from mild (grade 2) to severe (grade 4) based on the patient’s symptoms. Additionally, our findings indicate that allergy to spirulina is not commonly reported or diagnosed. However, most of the described cases (four of five) regarding allergy to spirulina according to WAO Anaphylaxis Guidance 2020 were classified as anaphylaxis. Furthermore, it is noteworthy that spirulina also possesses antiallergic properties, as evidenced by research studies. Our article delves into both the allergic and antiallergic potential of spirulina.

## 1. Introduction

New foods are sometimes introduced into our diet as supplements due to their nutritional properties [1], as part of a therapy, e.g., in a gluten-free diet [2], or as a key part of the provision of food for humanity in light of the growing world population [3]. Some foods are consumed in different parts of the world and discovered as new in other regions of the world. One example of a food newly discovered by the Western world is spirulina (*Arthrospira platensis)*. Several sources state that spirulina is a “novel food” [4,5,6]. The European Parliament uses this term to categorize any food or ingredient not used to a significant degree for human consumption within the EU before 15 May 1997 [7]. In practice, this means that, if a particular product was not widely used as food before that date, it can be classified as new or unknown in the context of human consumption safety and may be subject to special regulations regarding food safety. It is essential to ensure that new products are safe for consumers before introducing them to the market. Examples of this include *Bactris major*, *Cannabinoids*, *Xanthoparmelia scabrosa*, and zeolite. We can find *Arthrospira platensis* in the EU Novel Food Status Catalogue, but its description as of 13 March 2024 says that it is “NOT NOVEL IN FOOD—According to the information available to the Member States’ competent authorities, this product was used for human consumption to a significant degree within the Union before 15 May 1997. Thus, it is not considered to be ‘novel’ according to the provisions of the Novel Food Regulation (EU) 2015/2283 and its access to the market is not subject to the pre-market authorization in accordance with Regulation (EU) 2015/2283” [7]. Nevertheless, on the aforementioned page, we find the following: “However, other legislation may restrict the placing on the market of this product as a food in the EU or in some Member States. Therefore, it is recommended to check with the competent authority(ies) of the Member State(s)” [8]. Moreover, according to another document of the European Parliament’s Joint Research Centre by Araújo et al., 2021, we see that *Arthrospira platensis* is listed as a “Food supplement” in France [9]. This may suggest that it is considered to be a novel food in that country.

The commercial name of *Arthrospira platensis*—commonly called spirulina—originates from spiral filaments observed under a microscope [10]. It is sold as a dietary supplement in the form of compressed tablets, capsules, and powders. Due to its high protein and micronutrient content, spirulina is also added to snacks, pasta, cookies, and bread [11,12,13]. The high nutrient content of spirulina has multiple positive effects on human health; spirulina’s valuable properties have contributed to a growing interest in alga supplementation. Spirulina rose to prominence following its successful utilization by the US National Aeronautics and Space Administration (NASA) as a dietary adjunct for astronauts during space missions [10]. In 2019, spirulina (*Arthrospira* species) accounted for 99.5% of the total microalgae production, which reached 56,456 tons [14]. In the same year, the spirulina market was valued at USD 0.4 billion, and it keeps growing. In 2028, it is projected to reach USD 0.95 billion [15].

Spirulina is a trade name for dietary supplements containing oxygenic photosynthetic bacteria that grow best in subtropical alkaline lakes (pH 8.5–11.0) at temperatures of 35–39 °C [16,17]. Globally, this alga can be found in the wild in diverse areas: Lake Chad (Chad), Lake Texcoco (Mexico), and Lake Turkana (Kenya) [18]. Spirulina belongs to the family Phormidiaceae. The individual cells of *Arthrospira platensis*, a planktonic filamentous cyanobacterium, measure approximately 8 μm in diameter. The alga lacks a cellulose cell wall, which makes the process of preparation for ingestion straightforward as it does not require chemical or physical processing other than the simple yet effective process of sun drying [18]. Only a handful of regions around the world, such as Greece (Nigrita and Serres), Japan, India, the United States, and Spain, can provide the ideal sunny climate essential for production of these algae [19]. Spirulina is sold in the form of dried biomass of the genus *Arthrospira* species, characterized by a subtle marine scent that is more pronounced than the flavor [10,18]. The best-known of the 35 species of this photosynthetic Gram-negative class of cyanobacteria include Spirulina platensis (*Arthrospira platensis)*, Spirulina maxima (*Arthrospira maxima*), and Spirulina fusiformis (*Arthrospira fusiformis*) [18,20].

Spirulina platentis is one of most ancient algal species on our planet, with its origins tracing back approximately 3 billion years [21]. The remarkable history of spirulina species utilization dates back to the time of the Aztec civilization. One of the first records of spirulina usage dates from the Spanish conquest of Mexico in the 16th century. The present Western method of using these algae differs from that of the Aztecs. This civil Mesoamerican population used to gather the algae from Lake Texcoco and form them into bricks resembling blocks of modern cheese. On another continent, in central Africa, the ethnic group of Chad—the Kenebu tribe—collected spirulina from Lake Chad. The dried form of algae was incorporated into their indigenous cuisine in the form of dressing [18]. Spirulina’s immune-modulating functions and ability to alleviate various diseases have already been observed in the past, which is why it has been used in Chinese medical practice [21]. In addition to these beneficial effects on the immune system, allergic reactions to spirulina have been observed. In this review, we analyzed them and assessed their severity and collected information on the healing properties of spirulina for allergic diseases.

## 2. Benefits and Application of Spirulina

Spirulina owes its popularity to its high nutritional content. It is exceptionally rich in protein, which accounts for up to 60–70% of the dried weight, surpassing the protein content of meat and fish (15–25%) and soybean meal (35%) [18]. *Arthrospira platensis* contains an impressive number of eighteen amino acids, encompassing both essential and non-essential ones. Additionally, it serves as a valuable source of crucial amino acids, such as lysine, histidine, threonine, phenylalanine + tyrosine, valine, leucine, isoleucine, and methionine + cysteine [22]. The fat content ranges from 5 to 7 percent, with significant molecules including polyunsaturated fatty acids, such as gamma-linolenic acid and phenolic compounds. The carbohydrates, which comprise 10–15% of the dry weight, are branched polysaccharides structurally akin to glycogen [23]. The total nucleic acid content is less than 5% of the dry weight, which is lower compared with that of bacteria or yeasts, where it typically ranges from 4 to 10 percent [18,24] Additionally, spirulina is abundant in micronutrients, such as potassium, sodium, calcium, magnesium, iron, zinc, provitamin A (β-carotene), vitamin B1 (thiamine), B2 (riboflavin), B3 (nicotinamide), B6 (pyridoxine), B9 (folic acid), B12 (cyanocobalamin), vitamin C, vitamin D, and vitamin E (tocopherol). Consequently, vegetarians can benefit from spirulina as a significant source of B-group vitamins; therefore, spirulina may be of particular interest to people with a vegetarian diet. Spirulina contains vitamins B6 and B12, whose deficiencies are linked to depression—a condition that, according to the WHO, will be the most common disease in the world by 2030 [25].

The remarkably high micronutrient content of spirulina has drawn scientists’ attention, and the use of these algae extends to many fields of medicine, such as developing new treatments for various neurological diseases, obesity, or allergic diseases. Researchers have investigated spirulina and its potential therapeutic use in neurology. Several in vivo studies on animal models of Alzheimer’s disease [23,26] have demonstrated prevention of memory loss and reduction in oxidative stress damage [26]. Other studies have shown a positive impact of spirulina on Parkinson’s disease in animal models. Pabon et al. showed in a rat model that spirulina could have neuroprotective properties and could reduce microglial activation [27]. In a more recent study regarding a rat model of Parkinson’s disease, the researchers noticed improvements in locomotor activity and biomarkers of oxidative stress and inflammation [28]. Other scientists have investigated spirulina’s potential in treating obesity. Some clinical trials indicated that spirulina facilitates reduction in body weight in obese patients [29,30]. Yousefi et al. showed reductions in body weight, body mass index (BMI), serum total cholesterol levels, and appetite in patients after 12 weeks of daily treatment with 4 × 500 mg spirulina tablets daily along with a restricted-calorie diet (RCD) [29]. Spirulina has been scientifically validated for its positive impact on lipid metabolism. In their meta-analysis, Seban et al. suggested that spirulina supplementation results in a significant decrease in total cholesterol, low-density lipoprotein cholesterol (LDL-C), and triglyceride levels, as well as a significant increase in high-density lipoprotein cholesterol (HDL-C) [24]. Consequently, individuals afflicted with obesity could benefit from supplementing spirulina.

## 3. Methods

In order to find information about potential allergic reactions to spirulina in this literature review, we used PubMed and Scopus databases to search for papers related to spirulina allergy or hypersensitivity. The following components were included in the search string: synonyms, search filters, and Medical Subject Headings (MeSHs). Table 1 shows the list of the search terms and the number of studies published prior to July 2023.

This literature review was conducted using the Preferred Reporting Items for Systematic Reviews and Meta-Analysis (PRISMA) guidelines [31]. The aim of the selected papers was to answer the following research questions:What is the prevalence of allergic reactions to spirulina consumption?What is the severity of allergic reaction to spirulina consumption?What are the management strategies for individuals experiencing allergic reactions to spirulina?

Articles were screened regarding the title and abstract by two independent authors. Any disagreements were resolved through discussion (until a unanimous decision was made) or through consultation with a third author. In the next step, the articles were entered into Excel to remove duplicates. Following that step, records were classified and selected. We conducted screening based on the criteria of conforming to our review’s objective and the full text being available in English. Extracted data encompassed descriptive information about clinical and methodological factors, including age, gender, location, allergen source, criteria of diagnosis, clinical symptoms, and treatment. Out of 49 screened articles, 45 were excluded because they were irrelevant to the topic of the study, or their full text was not available in English. Ultimately, we analyzed 4 case studies with 5 patients who experienced an allergic reaction to spirulina. The process of study inclusion can be seen on Figure 1. Furthermore, we assessed patients’ symptoms of allergic reaction to spirulina using the World Allergy Organization (WAO) systemic allergic reaction grading system. The supervisor, M.K., was consulted to resolve any disagreements in classifying symptoms.

## 4. Results

An extensive literature search was performed to identify all the available cases of allergy to spirulina. Of the forty-nine identified records, only four case reports describing five patients were included for further analysis. The discovered descriptions of cases were in the form of case reports, letters, and brief communications or poster presentations. All the cases were reported in Europe (France, The Netherlands, Switzerland, and the UK). The data are summarized in Table 2.

An allergy to spirulina was reported for the first time in 2010. The patient was a 14-year-old male who developed urticaria, labial edema, and an asthma attack six hours after ingesting five spirulina tablets. The patient required treatment with second-generation antihistamine drugs and corticosteroids. The authors did not include any information on past medical history or other allergic reactions. Western blotting was used to identify the β-chain of C-phycocyanin (a cyanobacterial pigment) as a possible allergen for provoking an allergic reaction. The patient underwent oral challenge (OC) and a skin prick test (SPT). After oral administration of progressively increasing doses of spirulina, equivalent to four tablets, within a three-hour timeframe, the patient experienced diarrhea at hours 2 and 5 after the final challenge. Additionally, 30 min after the last episode of diarrhea, he developed diffuse erythema. The patient was diagnosed with an allergy to spirulina based on his symptoms, the positive skin prick test, and the oral challenge findings [32].

The second patient was identified in the Netherlands: a 17-year-old male who developed a systemic reaction 10 min after ingesting a 300 mg spirulina tablet. The symptoms included tingling of the lips, angioedema of the face, pruritic exanthema and urticaria of the arms and trunk, nausea, abdominal pain, wheezing, dyspnea, and inspiratory stridor. The patient required extensive emergency treatment, including intramuscular (im) epinephrine, intravenous (iv) antihistamines, iv glucocorticoids, and nebulized beta-2 agonists. His medical history included atopic dermatitis, asthma, allergic rhinitis (AR) (in response to grass, house dust mite, dogs, and cats), and a possible pollen-food syndrome (oral allergy symptoms in response to tomatoes and cucumbers). The diagnosis of spirulina allergy was based on clinical manifestations and a positive skin prick test with diluted *Arthrospira platensis* [33].

Another report from the UK documented two patients who had an allergic reaction to ingested spirulina powder in a smoothie. The symptoms were generalized itching, burning in the soles of the feet, shortness of breath, and urticaria. Patients 3 and 4 had a medical history of oral allergy syndrome, with a positive reaction to birch pollen diagnosed by component-resolved testing (rBet v 1), and seasonal rhinitis. Both patients described mild oral reactions following fresh fruit and vegetable consumption and their uncooked versions. Diagnosis of allergy to spirulina was based on symptoms and a positive skin prick test with spirulina powder. Unfortunately, this was a short description of spirulina allergy without specifying the symptoms in the individual patients [34].

The last and most recent case of an allergy to spirulina was reported in Switzerland in January 2022. The patient was a 48-year-old woman who experienced an episode of allergy to spirulina 3 and 7 h after ingesting three tablets of spirulina (400 mg each). The allergic reaction manifested as mild left plantar edema and acute tongue edema. The symptoms were successfully managed with antihistamines and corticosteroids in the emergency department. The fifth patient’s history included two episodes of pharyngeal edema that were successfully managed with intravenous antihistamines and corticosteroids. The patient did not have any atopic diseases, such as food or drug allergies. In December 2021, she underwent surgery without complications, and a minor wound was treated with iodopovidone. The patient underwent an oral challenge, which was positive (erythematous swelling in both heels and subjective “sensitization of the throat swelling” [sic]). The pharyngeal symptoms resolved 1 h after oral administration of an antihistamine and oral corticosteroids. However, plantar edema persisted for 24 h. Diagnosis was made based on the clinical symptoms of allergy, a positive SPT, OC, and basophil activation test (BAT). Positive BAT results suggest an IgE-mediated hypersensitivity mechanism with delayed clinical reactivity [35].

### Evaluation of the Severity of Analyzed Cases of Allergy Reaction to Spirulina

Allergic reactions may have various symptoms, from mild to severe, including anaphylaxis, which may even threaten the life of the allergic person. According to WAO Guidance 2020, anaphylaxis is defined as a severe and life-threatening systemic reaction characterized by rapid onset, with potentially life-threatening airway, breathing, or circulatory problems, and it is usually, although not always, associated with skin and mucosal changes. Multiple scales have been used to assess the severity of allergic reactions [36,37,38,39,40,41]. For this review, we decided to use the WAO systemic allergic reaction grading system published in Anaphylaxis Guidance 2020 since it is a recent scale that was reviewed and endorsed by multiple allergy societies associated with the WAO. According to the WAO classification, grades 3, 4, and 5 are categorized as anaphylaxis, and the lower grades are not classified as anaphylaxis [36]. All the symptoms that individual patients experienced after spirulina ingestion are shown in Table 2. We assessed the severity of the reaction after spirulina ingestion according to the WAO 2020 classification for each of the analyzed cases. Each symptom was assigned a corresponding severity according to the above scale. The patient’s most severe symptom determined the severity of the reaction in which the patient was classified. After a thorough analysis of the first patient’s symptoms (urticaria, labial edema, and asthma), we classified his reaction as grade 3. The symptoms of the second patient (tingling of the lips, angioedema of the face, itching exanthema, urticaria of arms and trunk, nausea, abdominal pain, wheezing, dyspnea, inspiratory and stridor) received grade 4 according to the scale above. Subsequently, the symptoms of both the third and fourth patient (generalized itching, burning in soles of feet, shortness of breath, and urticaria) were assigned a grade of 3. According to the WAO classification from 2020, the symptoms of the first four patients can be identified as anaphylaxis to spirulina. The last of the analyzed cases of allergic reaction to spirulina was IgE-mediated hypersensitivity, and we did not classify it as anaphylaxis [35]. Bear in mind that the patient experienced two grade 1 symptoms from two different systems (mild left plantar aspect swelling and acute tongue swelling), which we classified as grade 2 in accordance with the WAO scale. As an allergic reaction can occur after different foods consumed by a patient, a DBPCT should be performed to prove that spirulina is responsible for the hypersensitivity symptoms. However, only in two out of five patients was an oral provocation test performed, which is the “gold standard" in food allergy diagnosis. In the analyzed cases, there is no information on how these provocation tests were performed. In each of the cases in question, SPTs were performed, and in each case these tests were positive. A detailed analysis is shown in Table 3.

After evaluating the cases, we noticed that three of five patients required a hospital visit to effectively manage the symptoms they experienced after ingesting spirulina tablets or powder in a smoothie. All the patients were administered antihistamines and corticosteroids [32,33,34,35]. The patient who experienced the most severe allergic reaction (grade 4 according to the WAO scale) required an additional medical intervention, which included intramuscular administration of epinephrine together with nebulized beta-2 agonists, to alleviate the intensified symptoms [33]. In every case, diagnosis of allergy to spirulina was made based on clinical symptoms of allergy and a positive SPT [32,33,34,35]. In some patients, an oral challenge test (in two cases) and a BAT (in one case) were additionally used to confirm the diagnosis (Table 2) [33,35].

## 5. Discussion

### 5.1. Spirulina as an Allergy-Inducing Factor

To the best of our knowledge, this is the first review focused on the topic of allergy to spirulina. There was a short report on edible algal allergenicity published in 2023 [42]; however, in that study, the authors found only two cases of allergy to spirulina since their main aim was to assess general algal allergenicity.

By now, we know a great deal about the nutritional value of spirulina and its potential to treat some diseases. The diagnostic possibilities of IgE-mediated allergy based on component-resolved diagnosis (CRD) were discovered in recent years. Its application, together with the accumulated knowledge on allergenic proteins facilitating diagnosis, proved to be a key element in the understanding of allergic cross-reactions, hitherto not always pathogenetically understood [43]. However, little is known about the allergic proteins in spirulina. Petrus et al., 2010 identified β-chain c-phycocyanin as the allergen responsible for the first reported case of spirulina-induced anaphylaxis [32]. C-phycocyanin, which is blue, is the main pigment of spirulina [17]. According to in vitro studies, this protein has various biological effects, such as antioxidant, anti-inflammatory, antiplatelet, hepatoprotective, and cholesterol-lowering properties. Additionally, an in vivo study revealed that C-phycocyanin does not induce acute or subacute toxicity [44]. In a recent study, Bianco et al. demonstrated C-phycocyanin to be a spirulina allergen and proposed a new paradigm to search for allergenic proteins in novel foods by integrating proteomic analysis and in silico sequence homology prediction. Those authors explored the allergenicity of proteins found in spirulina extract following Food and Agriculture Organization (FAO) of the United Nations and World Health Organization (WHO) guidelines, in which cross-reactivity occurs when (i) the percentage of identity of amino acid sequences is higher than 35% using a window of 80 amino acids or (ii) there is an identity of at least six contiguous amino acids [4,45]. Six proteins detected in spirulina extracts exhibited significant sequence homology with one or more proteins from other organisms categorized as food allergens. The following proteins were identified in spirulina extracts: two thioredoxins (Uniprot code D4ZSU6 and K1VP15), superoxide dismutase (C3V3P3), glyceraldehyde-3-phosphate dehydrogenase (K1W168), and triosephosphate isomerase (D5A635). Putative proteins show sequence homology to food allergens related to pistachios, fish, shrimp, maize, and other edible products [4] which we have presented in the Figure 2.

The two thioredoxins, which were identified as a novel cross-reactive cereal allergen family, may contribute to the clinical manifestation of baker’s asthma [46]. One of these, Thioredoxin h1 (Q4W1F7), is an allergen found in corn seeds (Zea m 25) [6,46,47]. Other spirulina allergens are proteins with the structure of Glutathione-Dependent Peroxiredoxin (K1X048), which is related to fungi and can cause symptoms through skin and inhalation [4].

Protein C3V6P3 [48] of spirulina, a superoxide dismutase fragment, is associated with seven superoxide dismutases that are split into three groups of isoforms. The first group of isoforms (Q9FSJ2, P35017, and Q9STB5) is related to allergen Hev b 10 found in *Hevea brasilinesis*—a tree that is used to produce latex [49,50,51]. The second group of superoxide dismutase isoforms (B2BDZ8) is related to a food allergen from pistachio vera seeds: Pis v 4 [52]. The third group of isoforms (Q92450, M5ECN9, and Q873M4) is related to the fungi kingdom [4,53,54,55].

The other identified proteins were two glyceraldehyde-3-phosphate dehydrogenases. One of these molecules, C7C4X1, is related to baker’s asthma [56], while the second one, A0A5N5Q6M7, is associated with *Pangasianodon hypophthalmus* (striped catfish), categorized as allergen Pan h 13 [57].

Bianco et al. discovered that eight triosephosphate isomerases have relevant sequence homology with the last detected spirulina protein. The first two (L7UZA7 and A0A088SAX2) are associated with *Dermatophagoides farinae* [58,59], and protein Q9FS79 is associated with *Triticum aestivum*. These proteins are not considered to be food allergens but are inhalant allergens [4].

The rest of the identified triosephosphate isomerases are linked to food allergens of fish or crustaceans, A0A5N5Q6M9 to Pan h 8 from *Pangasianodon hypophthalmus* [60], A0A1L5YRA2 to Scy p 8 from *Scylla paramamosain* (mud crab) [61], D7F1Q0 to Cra c 8 from *Crangon crangon* (Brown shrimp) [4,62], F5A6E9 to Pro c 8 from *Procambarus clarkia* (Red swamp crayfish) [4,63], and B5DGL3 to Sal s 8 from *Salmo salar* (Atlantic salmon) [4,64].

As of 17 March 2024, none of the proteins from *Arthrospira platensis* have been included in the systematic allergen nomenclature approved by the World Health Organization and the International Union of Immunological Societies (WHO/IUIS) Allergen Nomenclature Sub-Committee. This nomenclature considers proteins with proven allergenic properties [65].

### 5.2. Spirulina—As an Allergy-Alleviating Factor

In our article, we meticulously analyzed the allergic potential of spirulina, striving to gather as much information as possible on this topic. Moreover, there are intriguing reports of attempts to utilize spirulina as a method for treating allergic diseases. The first report on spirulina’s antiallergic properties came from a 1997 study by Yang et al., who investigated the impact of spirulina platensis powder (SPP) on treating anaphylactic reactions in rats. In this experiment, Yang’s team utilized one of the most powerful secretagogues (compound 48/80) to stimulate mast cell degranulation that would cause a systemic anaphylactic shock in the animal model. Another essential molecule used in the experiment was anti-dinitrophenyl (DNP), which was used to generate passive cutaneous anaphylaxis. Evidence from that experiment indicated that SPP completely suppressed compound 48/80-induced anaphylactic shock at doses of 0.5 and 1.0 mg/g of body weight (BW). It also notably reduced serum histamine levels triggered by compound 48/80. At a dose of 0.5 mg/g of BW, SPP inhibited passive cutaneous anaphylaxis activated by anti-DNP IgE by 68.7%. Additionally, SPP demonstrated dose-dependent inhibition of histamine release from rat peritoneal mast cells (RPMCs) induced by compound 48/80. Furthermore, SPP exhibited a significant effect on anti-DNP IgE-induced histamine release or tumor necrosis factor-α production from RPMCs. These were the very first findings that suggested that SPP may contain compounds that effectively suppress mast cell degranulation in rats [21].

In 1998, another research team, Kim et al., showed the inhibitory effects of the microalgae on mast cell-mediated immediate-type allergic reactions in in vivo and in vitro studies. In rats, spirulina demonstrated a dose-dependent inhibition of systemic allergic reactions induced by compound 48/80. Remarkably, doses ranging from 100 to 1000 mg/g of BW, administered intraperitoneally, led to a 100% inhibition of compound 48/80-induced allergic reactions, as well as significantly curtailed local allergic reactions triggered by anti-DNP IgE. Furthermore, spirulina, at concentrations of 0.001 to 10 mg/mL, exhibited a dose-dependent inhibition of histamine release from RPMCs activated by compound 48/80 or anti-DNP IgE. When spirulina (10 mg/mL) was introduced, the level of cyclic AMP in RPMCs transiently and significantly rose approximately 70-fold at 10 s compared with that in the control group. Additionally, spirulina at a dose of 10 mg/mL displayed a significant inhibitory effect on anti-DNP IgE-induced tumor necrosis factor- α production [66].

Hayashi et al. (1998) demonstrated that dietary spirulina offers protective benefits in orally induced food allergy in mice by inhibiting the elevation of IgE levels, while simultaneously boosting IgA antibody levels to suppress allergic responses [67]. Spirulina neither induces nor exacerbates allergic reactions, such as IgE-mediated food allergies. Furthermore, when consumed whether alongside or before antigen exposure, this algal extract may substantially increase IgA antibody levels, potentially offering protection against allergic reactions. This statement was supported by Hayashi et al., who examined the production of antibodies, including IgA, IgE, and IgG1, in mice to investigate the potential protective effects of spirulina against food allergies and microbial infections. Mice orally immunized with crude shrimp extract as an antigen (Ag group) exhibited an increase in serum IgE antibody levels. However, treatment with spirulina extract (SpHW) did not further elevate IgE levels. Conversely, spirulina extract enhanced the IgG1 antibody levels, which were initially increased by antigen administration. Notably, spirulina extract significantly boosted the IgA antibody levels in intestinal contents when administered concurrently with a shrimp antigen, in comparison with the IgA levels in mice treated with the shrimp antigen alone. Additionally, spirulina extract increased IgA antibody production in the culture supernatant of lymphoid cells, particularly in the spleen and mesenteric lymph nodes of mice treated with spirulina extract for 4 weeks prior to antigen stimulation [47]. Four years later, Japanese researchers T. Hirahashi et al. identified the molecular mechanism underlying spirulina’s impact on the human immune system by examining the blood cells of volunteers before and after they received an oral dose of a hot water extract from Spirulina platensis. Administration of the microalga extracts increased interferon gamma (IFN-γ) production and damaged natural killer cells in male volunteers [68].

Another step in deepening our understanding regarding the role of spirulina in immune response modulation was a 2005 study by Chen et al., who demonstrated preventive and therapeutic effects of spirulina in rats with AR. That study showed that spirulina can lessen the symptoms of AR by demonstrating alleviation of inflammatory responses in the nasal mucosa of subjects receiving SPP. Particularly, the data illustrate a statistically significant decline in both the number of mast cells and the extent of their degranulation in the group treated with SPP in comparison with those parameters in the control group (*p* < 0.01). Moreover, the evidence shows a statistically significant reduction in serum histamine and total IgE levels in the SPP-treatment group compared with those levels in the control group (*p* < 0.01). Collectively, these findings imply the potential use of SPP in mitigating inflammatory responses in the nasal mucosa of patients with AR [69].

Also in 2005, Mao et al. published the first human study revealing the positive impact of spirulina on AR patients. The authors of that study evaluated the effectiveness of spirulina by assessing the production of crucial cytokines (interleukin [IL]-4, IFN-γ, and IL-2) involved in regulating IgE-mediated allergy. In this randomized double-blind placebo-controlled crossover trial, allergic individuals ingested either placebo or spirulina daily, at doses of 1000 mg or 2000 mg, for 12 weeks. Peripheral blood mononuclear cells obtained before and after spirulina intake were stimulated with phytohemagglutinin (PHA), and the cytokine levels in the cell culture supernatants were measured. While spirulina did not appear to affect the secretion of Th1 cytokines (IFN-γ and IL-2), it significantly decreased (by 32%) the IL-4 levels from PHA-stimulated cells when administered at a dose of 2000 mg/day. Based on this evidence, the authors suggested that spirulina can influence the T helper profile in AR patients by extinguishing the differentiation of Th2 cells, partly through inhibiting IL-4 production [70].

In a 2008, a randomized controlled trial (RCT) by Cingi et al. reported that a daily intake of five spirulina tablets (2000 mg/day) for 6 months revealed significant efficacy compared with a placebo (*p* < 0.001 ***, observed power = 100%) in alleviating AR symptoms, such as nasal discharge, sneezing, nasal congestion, and itching [71]. In a recent RCT conducted in 2020, a group of patients receiving 2000 mg of spirulina per day was compared with the control group treated with 10 mg of cetirizine for treatment of AR during a 2-month period. After this intervention, the rates of rhinorrhea (*p* = 0.021), nasal obstruction (*p* = 0.039), and smell reduction (*p* = 0.030) in the spirulina group improved significantly compared with those in the cetirizine group. Additionally, sleep quality, daily working, and social activity also improved significantly in the spirulina group (*p* < 0.05). One of the conclusions reached by the authors of this study was that “The use of spirulina seems to be more effective than the administration of cetirizine in improving of both clinical presentations and inflammatory mediators of AR patients”, and the second conclusion was that “Spirulina should be considered as an alternative treatment in patients with AR” [72].

## 6. Limitations

In the course of writing this review, we have encountered various challenges. The reported cases describing allergic reactions were not full articles but only in the form of short case reports, letters, poster presentations, or brief communications. The literature on this topic suffers from inconsistent clinical data and a limited sample size of just five patients. The information provided in the studies lacked sufficient detail, leaving us with many unanswered questions about allergic reactions to spirulina and allergens, which could be responsible reactions. The analyzed records included symptoms that were challenging to classify, for example, “mild left plantar aspect swelling” or “burning in the soles of feet”.

Symptoms of allergic reactions in individual patients after consuming spirulina were analyzed according to Anaphylaxis Guidance 2020, which is one of the most recent scales, and it is accepted by multiple scientific societies. Despite a long history of attempts at grading anaphylaxis and a number of existing scales that are constantly being improved, debate continues as to which scale is the most suitable grading system of allergic reactions. The discrepancies between individual scales are due to diverse settings, age groups, and triggers, such as anesthesia, food, or venom. One of the limitations of our article is that we assessed allergenicity based on available literature data, but we did not evaluate the structural composition of individual proteins. Additionally, in our review, we did not assess the reactivity of sera to specific spirulina proteins. Further research is necessary to demonstrate or confirm the allergenicity of specific spirulina proteins. We must also note that perhaps the low rate of allergic reactions to spirulina is related to the limited amount of spirulina consumed worldwide. Moreover, it cannot be ruled out that, as its consumption increases, the number of allergic reactions to *Arthrospira platensis* may increase.

## 7. Conclusions

Due to the health-promoting properties of spirulina, its consumption has undergone a notable increase in recent years. This raised questions pertaining to the safety of products containing spirulina and led to heightened scrutiny of these algae. Despite spirulina’s dominance in global microalgae production, our literature review suggests that spirulina allergy is not commonly reported or diagnosed. Nevertheless, a significant proportion of documented cases (four of five) of spirulina allergy, as per WAO Anaphylaxis Guidance 2020, were classified as anaphylactic reactions. Despite this observation, there is a notable absence of epidemiological studies on hypersensitivity to spirulina. Consequently, it is imperative to conduct further research to evaluate the risks of hypersensitivity and cross-reactivity associated with spirulina.

## Figures and Tables

**Figure 1 foods-13-01052-f001:**
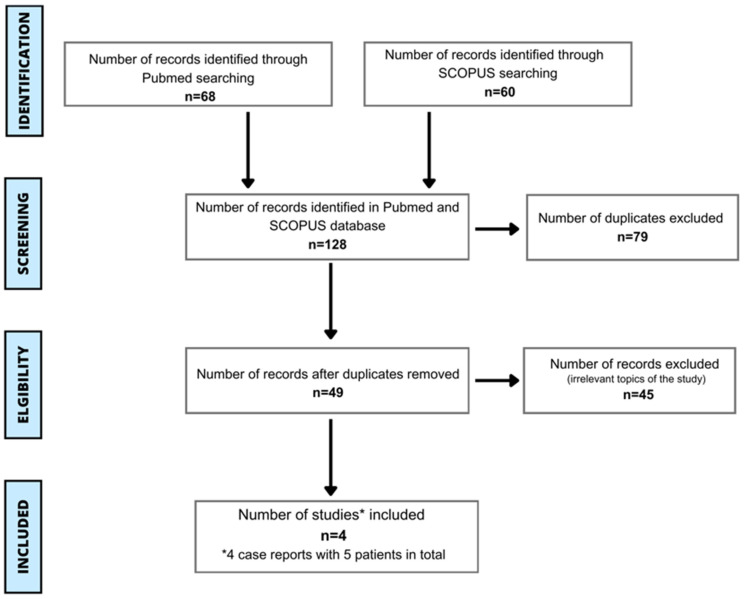
PRISMA flow chart displaying the process of study inclusion [31]. * 4 case reports with 5 patients in total.

**Figure 2 foods-13-01052-f002:**
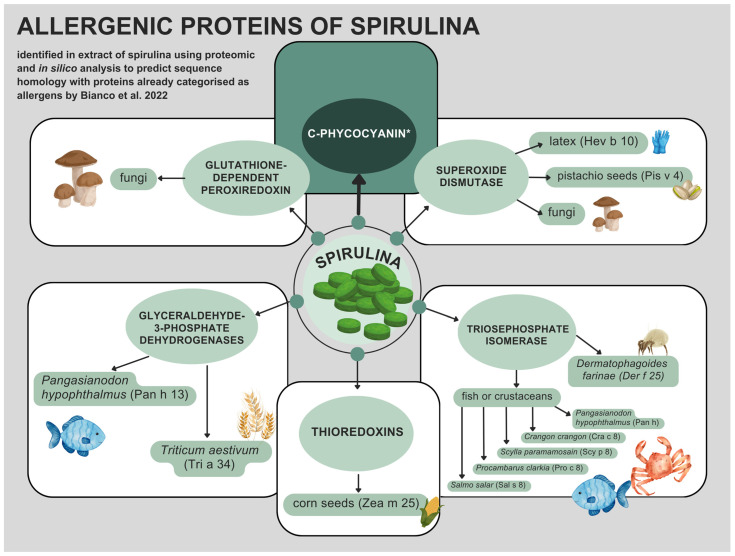
Presentation of the six proteins identified in a 2022 study by Bianco et al. using proteomic and in silico analyses following WHO/FAO guidelines shows significant homology with proteins already categorized as allergens in other allergen sources [6]. The protein C-phycocyanin, identified as a spirulina allergen that triggered the first case of anaphylaxis reported in the literature in 2010 [32], was subsequently validated as a spirulina allergen by Bianco et al. in 2022 [4].

**Table 1 foods-13-01052-t001:** The terms used to search the databases and the number of publications related to allergy to spirulina found in PubMed and Scopus before July 2023.

Search	Search Term	PubMed	Scopus
1	(allergy) AND (spirulina) AND (human)	19	30
2	(allergy) AND (*Arthrospira*) AND (human)	19	5
3	(hypersensitivity) AND (spirulina) AND (human)	13	15
4	(hypersensitivity) AND (*Arthrospira*) AND (human)	13	2
5	(anaphylaxis) AND (spirulina) AND (human)	2	5
6	(anaphylaxis) AND (*Arthrospira*) AND (human)	2	3
Summary	Total = 128	68	60

**Table 2 foods-13-01052-t002:** Summary of case reports of allergies to spirulina (*Arthrospira platensis*) found prior to July 2023.

Patient	Author(s), Kind of Article, Year	Location	Age and Sex	Form of Ingested Spirulina	Diagnosis of FA/FH	Clinical Symptoms	Treatment
1	Petrus et al. [32]; Case report, 2010	France	14 y/o male	tablets	Clinical symptoms of allergy, positive SPT and OC	urticaria, labial edema, asthma	2nd-generation antihistamines, corticosteroids
2	Le et al. [33]; Short communication, 2014	The Netherlands	17 y/o male	tablets	Clinical symptoms of allergy, positive SPT	tingling of the lips, angioedema of the face, an itching exanthema, urticaria of arms and trunk, nausea, abdominal pain, wheezing, dyspnea, inspiratory stridor	im ephinephrine, iv antihistamines, iv glucocorticoids, nebulized beta-2 agonists
3 and 4	Pimblett [34]; Poster, 2020	The United Kingdom	2 patients (ND about age and sex)	powder	Clinical symptoms of allergy, positive SPT	generalized itching, burning in soles of feet, shortness of breath, urticaria	No data
5	Pescosolido et al. [35]; Letter, 2022	Switzerland	48 y/o female	tablets	Clinical symptoms of allergy, positive SPT, OC, BAT	mild left plantar aspect swelling, acute tongue swelling	iv antihistamines, iv glucocorticoids

FA, food allergy; FH, food hypersensitivity; SPT, skin prick test; OC, oral challenge; BAT, basophil activation test; im, intramuscular; iv, intravenous; y/o, years old.

**Table 3 foods-13-01052-t003:** Severity assessment of patients’ allergic reactions to spirulina described in the literature before July 2023, with the use of the scale published in World Allergy Organization Guidance 2020 [36].

Patient	Anaphylaxis	Grade	Symptoms		Assessment of Reaction Severity
1 [32]	No	1	Urticaria		3
			Labial edema		
		2	-		
	Yes	3	Asthma		
		4	-		
		5	-		
2 [33]	No	1	An itching exanthema		4
			Angioedema of the face		
			Nausea		
			Tingling of the lips		
			Urticaria of arms and trunk		
		2	Abdominal pain		
	Yes	3	Dyspnea		
			Wheezing		
		4	Inspiratory stridor		
		5	-		
3 and 4 [34]	No	1	Burning in soles of feet		3
			Generalized itching		
			Urticaria		
		2	-		
	Yes	3	Shortness of breath		
		4	-		
		5	-		
5 [35]	No	1	Acute tongue swelling	2 symptoms from different systems are characterized as grade 2	2
			Mild left plantar aspect swelling		
		2			
	Yes	3	-		
		4	-		
		5	-		

Gray shading of field shows information based on which we assigned patients’ symptoms with adequate grade of severity of allergic reaction. All symptoms are described exactly as in the literature.

## Data Availability

No new data were created or analyzed in this study. Data sharing is not applicable to this article.

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
