# Peer review of "Spirulina (Arthrospira platensis): Antiallergic Agent or Hidden Allergen? A Literature Review"

_foods, 2024, doi:10.3390/foods13071052_

Round 1

Reviewer 1 Report

Comments and Suggestions for Authors

The present manuscript focuses on the identification of literature reporting adverse reactions to spirulina. The authors further identifies the severity of the reported symptoms according to WAO Anaphylaxis Guidance (2020). As the awareness about the safety of novel foods is rising, the topic of this manuscript is relevant, since the potential to induce allergic reactions needs to be carefully assessed. However, there are very few articles reporting the occurrence of adverse reactions after spirulina ingestion. Because of that, the authors focused a great part of their discussion talking about the use of spirulina as a potential treatment for allergy, which is not the main objective of this work. Additionally, some parts of the manuscript are confused and the language is not appropriate. I suggest to the authors to substantially improve the manuscript, especially the introduction and discussion section. The main objective of the work needs to be carefully evaluated and reformulated. Only after a substantial improvement of the manuscript, its publication can be reconsidered.

Specific points:

Lines 35-39. The groups of foods defined as novel foods is not clear in this paragraph as well as in which group spirulina is included.

Line 81. Which plant-based proteins?

Figure 2 has low resolution and most of the information cannot be read.

Lines 423-426. This sentence make no sense here. Maybe it is better place it in the introduction.

Comments on the Quality of English Language

The English language needs to be carefully revised. Some parts are confused, making it difficult for the reader to understand.

Author Response

We would like to thank the Reviewer for their valuable help in revising our manuscript, leading to significant enhancements. All detailed answers are in the attached file.

Reviewer 2 Report

Comments and Suggestions for Authors

Major comments

The manuscript by Gromek et al. describes an interesting review about the importance of the microalgae Spirulina as an alternative for protein enrichment of foods. Moreover, the authors describe other immunological properties associated to its consumption. Undoubtedly, this reviewer agrees with this concept and with the identification of alternative additives containing proteins and understand the potential allergenicity. However, according to the title of the manuscript, the authors should focus the revision on allergenic reactions, but the information included in this sense is poor. This author recognises that there is little information about it. Only some studies are included, with poor results, lack of consistent clinical data, 5 patients, limited information about allergens... additionally, the studies considered are poster presentations and not published in peer review article (Please, review the literature and include valid references). In fact, no allergen is included in the Allergen Nomenclature database (allergen.org) which is the official place where only proteins with demonstrated effect as allergens are included.  Apart from that, the authors include other properties of the microalgae. In conclusion, this reviewer recommends to modify the title of the document and used a general one describing Spirulina as an alternative for protein enrichment and describe allergenicity in the manuscript. The authors should discuss that the allergenicity of the Spirulina is limited probably because of the consumption. Moreover, this reviewer misses in the discussion section the hypothesis about a potential spread of allergy in case of worldwide consumption. The model of kiwi is a perfect example. What is going to happen if Spirulina is highly consumed?  

Regarding the severity of the allergic symptoms, the authors should explain in detail this classification and how the authors describe it. This classification cannot be done without oral provocation test, specially because microalgae is generally consumed with other foods.

Minor comments

-          In general, the discussion is focussed on many different issues and the analysis about allergy is limited.

-          Lane 273. Proteins identify in Spirulina extract. Just these allergens? Cross-reactivity with cereals? What is the homology with cereals? The authors should discus in detail. The allergens mentioned are minor allergens. Figure 2 should be completed with immunoblots in order to suggest the IgE recognition. Please, describe homology and structure.

-          Table 3 is not needed.

-          References should be updated and include only peer-review articles.

-          Lane 68-69. This is a scientific article and historical opinions and adjectives should be avoided

-          Lanes 103-104. The authors should include that these results are not demonstrated in DBPCT

Comments on the Quality of English Language

NA

Author Response

(The authors gave the same response as above.)

Reviewer 3 Report

Comments and Suggestions for Authors

The paper "Unveiling Allergic Risks in Novel Foods: A Literature Review 2 on Allergy to Spirulina (Arthrospira platensis)" provides an overview of the literature regarding the allergenic properties of Spirulina, a popular novel food that has been incorporated into the human diet. In addition to spirulina's nutritional value and health benefits, the authors talk about the safety concerns related to allergy reactions to spirulina that have been documented in the literature.

The manuscript discusses the health advantages and nutritional value of spirulina, but it also highlights the possibility of serious allergic reactions, including anaphylaxis, which are still uncommon. 

There are some issues which has to be improved.

Line 34: Please be more precise in the statement „The European Parliament uses this term to categorize any food or ingredient not used to a significant degree for human consumption“ What is considered a significant degree?

Line 210: Please quote references as follows: [34-39].

Line 332: Please check the statement.

Line 243: The statement regarding the confusion among clinicians should be removed because it presents certain ethical and professional concerns and is not pertinent to the readers of the Journal.

Line 256: Please avoid the „hitherto not always pathogenetically understood“ as it is related to molecular cross-reactivity due to the similarity in three-dimensional structure.

 Line 294: Please correct the quotation of the references: [49-51].

Line 317: „Significan majority of patients" is not the best term for the relatively small number of patients the manuscript analyzes. So please refine the statement.

Line 389: Please use mast cells not mastocytes.

Line 418: „Therefore, the microalgae proved to be more effective than the antihistamine drug.” Perhaps, the statement should be given with some reserve as it is a quotation of the published results.

Author Response

(The authors gave the same response as above.)

Round 2

Reviewer 1 Report

Comments and Suggestions for Authors

The authors made the requested amendments.

The manuscript have now all the conditions to be published.

Reviewer 2 Report

Comments and Suggestions for Authors

The manuscript is really interesting and it has been significantly improved. Congratulations